# Genome-Wide Association Study of Feed Efficiency Related Traits in Ducks

**DOI:** 10.3390/ani12121532

**Published:** 2022-06-13

**Authors:** Qixin Guo, Lan Huang, Yong Jiang, Zhixiu Wang, Yulin Bi, Guohong Chen, Hao Bai, Guobin Chang

**Affiliations:** 1College of Animal Science and Technology, Yangzhou University, Yangzhou 225009, China; dx120190114@yzu.edu.cn (Q.G.); dx120200133@stu.yzu.edu.cn (L.H.); jiangyong@yzu.edu.cn (Y.J.); wangzx@yzu.edu.cn (Z.W.); ylbi@yzu.edu.cn (Y.B.); ghchen2019@yzu.edu.cn (G.C.); 2Joint International Research Laboratory of Agriculture and Agri-Product Safety, The Ministry of Education of China, Yangzhou University, Yangzhou 225009, China

**Keywords:** feed efficiency, phenotypic traits, genome-wide association, single nucleotide polymorphism study

## Abstract

**Simple Summary:**

Good feed efficiency (FE) is an important trait to ensure the economic output of the livestock and poultry industries. Herein, a genome-wide association study was conducted to identify potential variants and genes associated with seven FE measures in ducks. Genomic DNA samples of 308 ducks were collected and sequenced. All animals were evaluated concerning body weight gain (BWG), feed intake (FI), residual feed intake (RFI), feed conversion ratio (FCR), and weight at 21 (BW21) and 42 days of age (BW42). Overall, 4 (FCR), 3 (FI), 36 (RFI), 6 (BWG), 8 (BW21), and 10 (BW42) single nucleotide polymorphisms (SNPs) were significantly associated with these FE traits, respectively. Moreover, candidate genes close to the identified variants were found to be mainly involved in key pathways and terms related to metabolism. In summary, these findings improve our understanding of poultry genetics and provide new foundations for breeding programs aimed at maximizing the economic potential of duck breeding and farming.

**Abstract:**

Feed efficiency (FE) is the most important economic trait in the poultry and livestock industry. Thus, genetic improvement of FE may result in a considerable reduction of the cost and energy burdens. As genome-wide association studies (GWASs) can help identify candidate variants influencing FE, the present study aimed to analyze the phenotypic correlation and identify candidate variants of the seven FE traits in ducks. All traits were found to have significant positive correlations with varying degrees. In particular, residual feed intake presented correlation coefficients of 0.61, 0.54, and 0.13 with feed conversion ratio, and feed intake, respectively. Furthermore, data from seven FE-related GWAS revealed 4 (FCR), 3 (FI), 36 (RFI), 6 (BWG), 8 (BW21), and 10 (BW42) SNPs were significantly associated with body weight gain, feed intake, residual feed intake, feed conversion ratio, and weight at 21 and 42 days, respectively. Candidate SNPs of seven FE trait-related genes were involved in galactose metabolism, starch, propanoate metabolism, sucrose metabolism and etc. Taken together, these findings provide insight into the genetic mechanisms and genes involved in FE-related traits in ducks. However, further investigations are warranted to further validate these findings.

## 1. Introduction

Feed efficiency (FE) is an important trait that is often related to measures of animal productivity [1,2,3]. In the poultry industry, FE represents its competitive position against other animal protein sources and it can be regulated to effectively increase profit margins [4]. The traditional measures of FE, such as the ratio of feed consumed to observed body weight (BW) (i.e., feed conversion ratio (FCR)), can be used to successfully select animals that can achieve higher growth rates in livestock and poultry. To compensate for the shortcomings of FCR calculations, residual feed intake (RFI) has attracted significant attention as an alternative measure for FE assessment, being used as a production performance evaluation index for poultry since the 1970s and as a measure of feed utilization efficiency index of livestock in the 1960s [5,6,7]. RFI is defined as the difference between the actual animal feed intake (FI) in relation to its growth rate and BW during a specific feeding period, and it can accurately reflect the metabolic differences among individuals, in which metabolic differences are determined by genetic background [8,9]. Since individuals with high RFI have higher feed intake than those with a low RFI, using RFI as a negative selection trait is more likely to produce populations with low feed intake and high productivity [10,11]. In sheep, the genetic correlation between daily feed intake (DFI) and RFI was 0.61 [12]. In addition, the genetic correlation was in the range of 0.693 for China’s local chicken breed. The phenotypic and genetic correlations between feed intake, body weight, and ADG were all positive and within the moderate to high range with genetic correlations ranging from 0.28 to 0.67. The genetic correlation between feed intake and RFI (0.62) was positive but the correlation between feed intake and FCR was approximately zero in turkey [13]. In addition, other reports on turkey suggested the genetic correlation between DFI and RFI was 0.62 [13]. Hence, RFI is an effective indicator of feed utilization efficiency in livestock and poultry [6,7,14,15], thereby representing a valuable tool for genetic improvement of energy metabolism in non-fast-growing livestock and poultry populations.

Advances in diet formulations have significantly improved the FE for poultry production [14]. However, with increasing feed costs, further improvement by genetic and breeding strategies has become a particularly important aspect in the past decades. By integrating statistical genetics, molecular biology, and sequencing technology in numerous studies, the genetic determinants for many economic traits of farm animals have been revealed, such as the eggshell structure, glycogen content of skeletal muscle, body size, weight, and reproductive traits [15,16,17,18,19,20]. Genome-wide association studies (GWASs) have attracted significant attention to investigate the genetic architecture of phenotypic traits given the increasing availability of whole-genome sequencing data [21]. GWASs are an efficient approach used to screen and identify candidate genes and variants associated with traits of interest and diseases [22,23]. In poultry, great progress has been made concerning growth traits through genetic selection. Indeed, a 50–60% increase in growth rate has been attributed to genetic selection. Phenotypic and genetic selection for FE can have significant effects on carcass composition in pigs, cattle, and chickens [1,2,8]; however, information describing candidate variants associated with RFI and other traits is still lacking. Therefore, the present study aimed to investigate candidate markers associated with FI, RFI, FCR, body weight (BW) gain, and BW in ducks, and better understand their underlying contributions to FE regulation.

## 2. Materials and Methods

### 2.1. Ethical Approval

All experiments with ducks were performed in accordance with the Regulations on the Administration of Experimental Animals issued by the Ministry of Science and Technology (Beijing, China) in 1988 (last modified in 2001). Experimental protocols were approved by the Animal Care and Use Committee of Yangzhou University (YZUDWSY2017-11-07). All efforts were made to minimize animal discomfort and suffering.

### 2.2. Samples Collection and Phenotype Registry

A total of 308 F_2_ segregating populations were used for the GWAS. To construct the F_2_ segregating population, the F_1_ generation was produced from orthogonal crosses between 86 Chinese Crested and 13 Cherry Valley ducks that were randomly selected and divided into seven families. The number of offspring in the F_1_ generation exceeded 500 individuals, none of which had crest traits, and the male:female ratio was consistent. The F2 generation was produced from the natural mating of F_1_ hybrids and mating was internally limited to orthogonal experiments. The families were established while considering the following principles: (1) a male:female ratio of 1:3, (2) males and females in the same family were not from the same nest, and (3) female ducks within a family were not half-siblings. To avoid half siblings, we designed a mating system. The F_2_ generation was composed of almost 2000 ducks that displayed segregation of various genetic characteristics, including meat quality-related traits. When the ducklings hatched, they were weighed each week afterward. Three weeks after hatching, all members of the F_2_ generation were moved from the duckling house to a designed individual shed and raised to the age of 6 weeks. A slaughter experiment was performed with more than 800 ducks and several traits were measured, including meat quality. In all families, the color (white and black) trait followed the recessive inheritance of Mendel’s law of separation.

To obtain the RFI data, the BW of each duck at 42 and 21 days of age was measured. The FCR was calculated as the ratio of the FI using the grDevies R package. Briefly, after removing outliers (values greater than three standard deviations from the mean), the random remaining 308 ducks were used to calculate RFI as the residuals from a regression model of FI on metabolic BW. Finally, RFI was calculated using the previously described formula [24] (Equation (1))
(1)$$RFI=FI-\left(a+b_{1}\times BW_{21}^{0.75}+b_{2}\times BWG\right),$$
where $BW_{21}^{0.75}$ is the metabolic weight at 21 days of age, *BWG* is the weight gain from 21 to 42 days of age, *a* is the intercept, and $b_{1}$ and $b_{2}$ are partial regression coefficients of FI on $BW_{21}^{0.75}$ and *BWG*, respectively.

### 2.3. Phenotype Data Correlation Analysis

To help identify candidate variants, a normal distribution test was performed on all phenotypic data using the stats R package. The quickcor function in the ggcor R package (https://github.com/hannet91/ggcor, access on 1 February 2022) was used to perform correlation analysis between all traits evaluated. The phenotypes that did not follow a normal distribution were transformed using the Johnson method implemented using the Johnson R package [25]. Then, the transformed data were used in the subsequent genetic analyses.

### 2.4. Genomic DNA Isolation and Sequencing

To isolate high-quality genomic DNA, blood samples from F_2_ ducks were obtained for DNA extraction and biochemical examination. High-quality genomic DNA was extracted using the standard phenol-chloroform protocol. The purity and concentration of the samples were measured using a NanoDrop ND-2000 spectrophotometer (Thermo Fisher Scientific, Waltham, MA, USA) and agarose gel electrophoresis, respectively. The final concentration was adjusted to 50 ng/μL, and samples with an A260/280 ratio of 1.8–2.0 were finally submitted for sequencing. Two paired-end sequencing libraries with insert sizes of 350 bp were constructed according to the Illumina protocol (Illumina, San Diego, CA, USA). All libraries were sequenced using the Illumina NovaSeq platform, with an average clean read sequence coverage of 11.60.

### 2.5. Variant Calling, Filtering, and Data Analysis

A total of 308 sequences were aligned to the mallard genome using the Burrows-Wheeler Aligner (BWA) software (settings: mem−t 4 −k 32−M−R) [26]. The sample alignment rates were 96–98%. The average coverage depth for the reference genome (excluding the N region) was between 9.34–15.74×, and the 4× base coverage (≥4) was greater than 82.64%. The average coverage depths for the reference genome (excluding the N region) were 6.00× and 17.66×. Variant calling was performed for all samples using the Genome Analysis Toolkit (GATK) (version 3.7, Broad Institute, Cambridge, MA, USA; https://gatk.broadinstitute.org/hc/en-us, access on 10 February 2022) with the UnifiedGenotyper method. SNPs were filtered using the Perl script [27]. After filtering, the sequencing data retained 12.6 Mb of SNPs (filter conditions: only two alleles; single sample quality = 5; single sample depth: 5–75; total sample quality = 20; total sample depth: 308–1,000,000; maximum missing rate of individuals and site = 0.1; minor allele frequency = 0.05).

### 2.6. Linkage Disequilibrium Decay Analysis and Principal Component Analysis

Linkage disequilibrium (LD) among the markers was calculated for all SNPs using PLINK [28] (https://zzz.bwh.harvard.edu/plink/, access on 20 February 2022). The window size for LD calculation was set based on the number of SNPs located in each genome. Pairwise LD was determined using squared allele frequency correlations, and assessed by calculating *r^2^* for pairs of SNP loci. Principal component analysis (PCA) was conducted using PLINK with the 12,201,978 variants of the 308 ducks to estimate the population structure. The ggplot2 R package [29] was used for the visual analysis of the results.

### 2.7. GWAS Analysis

The multilocus linear mixed model of fixed and random model circulating probability unification (FarmCPU) method was used to conduct the association analysis between the SNPs and FE traits [30]. The FarmCPU method used iterative fixed and random effect models to perform the GWAS, and it was able to minimize confounding between the testing markers and kinship. The fixed-effect model included numerous pseudo-quantitative trait nucleotides (QTNs) as covariates to control for false positives and test markers one at a time. In the fixed-effect model, possible association markers were generated in each round, and in the random-effect model, the Settlement of MLM Under Progressively Exclusive Relationship (SUPER) algorithm selected pseudo-QTNs among the possible association markers. To overcome the overfitting problem of the fixed-effect model, pseudo-QTNs were used to define individual kinships. To reduce the effect of population stratification on false positives, the first five principal components were employed as covariate variables in the GWAS models. To calculate the adjusted or transformed phenotype data of the traits, the fixed effect model was as follows (Equation (2)):(2)$$y_{adj}=Xb_{X}+M_{t}b_{t}+S_{j}d_{j}+e,$$
where $y_{adj}$ is the vector of adjusted or transformed phenotype data of the traits; X is a fixed-effects matrix constructed by the five highest principal components; $M_{t}$ is the matrix of t pseudo-QTN genotypes, initiated as an empty set; $b_{X}$ and $b_{t}$ are the respective effects of the three principal components and *t* pseudo-QTNs, respectively; $S_{j}$ is the genotype of the marker; $d_{j}$ is the impact of the marker; and *e* is a vector of random residuals. A random-effect model was updated by utilizing the SUPER algorithm to choose pseudo-QTNs in three steps: (i) the SNPs were sorted by their *p*-values determined for one characteristic; (ii) the SNP with the lowest *p*-value was selected as the sample for each chromosomal bin. To create kinship, the most important bins were selected. To optimize the constrained maximum probability for a trait, the size and number of bins used were considered as parameters. For the subsequent association test, the selected SNPs (each representing a bin) were used as the basis for the SNP pool. As a result, SNPs that were in LD (*r^2^* > 0.8) with the tested SNP were not included in the SNP pool. The random-effects model was set as follows (Equation (3)):(3)y=u+e

N (0, K^2^_u_), indicating that u is a genetic effect of the individual. The pseudo QTNs used to make this matrix were called “kinship matrixes,” and this matrix is called “kinship matrix” because it is made up of kinship matrices that look like real kinship matrices. SNP genotypes were coded as 0, 1, and 2, which were changed using PLINK.

The Bonferroni correction method was used to set the significance threshold for selecting important SNPs. To maintain the type I errors at 5%, the genome-wide significance threshold was set at 4.097696 × 10^−9^ (0.05/12201978), and the rate was maintained at that level.

### 2.8. Gene Ontology (GO) and Kyoto Encyclopaedia of Genes and Genomes (KEGG) Analyses

Based on the LD attenuation distance calculated by PopLDdecay [31], annotation of related genes in a certain region upstream and downstream of the physical location of significant SNPs was performed. The sequences of the relevant genes were extracted from the mallard genome and translated into a protein sequence, which was then submitted to KOBAS 3.0 [32]. Chicken was selected as the reference species, and the hypergeometric test and Fisher’s exact test were used as statistical methods.

## 3. Results

### 3.1. Basic Descriptive Statistics of FE

Descriptive characterization of the 308 ducks concerning BW at 42 and 21 days of age, FCR, FI, BWG, and RFI is provided in Table 1. Overall, the minimum and maximum values of RFI were −924.352 and 941.7816 g/d, respectively. The coefficient of variation of the FCR (12.1329%) was higher than that of the FI (13.897%). Distribution analysis of the seven FE traits showed that they did not fit the normal distribution (Appendix A). Thus, Johnson transformation was performed to ensure that all traits fitted a normal distribution pattern (Appendix A). In addition, Pearson correlation analysis showed significant positive correlations with varying degrees among all traits. The correlation coefficients between RFI and FCR and FI were 0.61 and 0.54, respectively. (Figure 1).

### 3.2. SNP Distribution, Population Structure, and LD Decay

Among all 308 ducks evaluated, a total of 12,201,978 SNPs with minor allele frequency > 0.05 and maximum missing rate < 0.1 were finally obtained and used for the subsequent analyses. All filtered SNPs, with an average density of 10,574.9207 SNPs/Mb, were distributed within the 40 autosomal chromosomes and chromosome Z (Figure 2a). PCA revealed two potential subpopulations, which indicated that population stratification existed in our genomic samples (Figure 2b). Nevertheless, this stratification had little effect on the phenotypes evaluated, since individuals with different traits were evenly distributed between the two clusters. In addition, the maximum LD was found to be 0.582 and the LD of the half-maximum LD was 0.291. The threshold for useful LD was set at *r^2^* = 0.1 at distances of 80,977 (Figure 2c).

### 3.3. GWAS for Traits of FE

Next, candidate SNPs related to the seven FE traits in ducks were investigated within the initially detected 12,201,978 SNPs. GWAS results showed that six SNPs located on chromosomes 4, 7, 16, 22, and 25, which were close to *TBC1*, *INPP5A*, *STK32C*, *PRKG1*, *WFDC8*, *RPL37*, *ROCK2*, *HOXB3*, and *HOXB2*, were associated with BWG (Appendix A). The four SNPs found nearest to *EDIL3*, *COX7C*, and *ERCC4*, located on chromosomes 15 and Z, were significantly associated with FCR (Figure 3, Appendix A). Three SNPs near *OTOL1*, *SI*, which is located on chromosome 9, were found to be related to FI (Figure 3, Appendix A). In addition, 36 SNPs located on chromosomes 1, 3, 4, 8, and 17, which were near *LSAMP*, *GAP43*, *B3GNT8*, *ENSAPLG00020009332*, *FAM241A*, *RAP1GDS1*, *UNC5C*, *SUCO*, *ENSAPLG00020001335*, *ENSAPLG00020001341*, and *GSTT1*, were found to be candidate SNPs of RFI in ducks (Figure 3, Appendix A). Eight SNPs located on chromosomes 1, 5, 7, 9, 11, 13, and 25, and close to *CCDC59*, *PPFIA2*, *TBC1D4*, *SUSD6*, *PNLIPRP2*, *N**GEF*, *GABRE*, *ENSAPLG00020002037*, *CBLN1*, and *MRPL10*, were candidates associated with weight at 21 days of age (Appendix A). Lastly, 10 SNPs located on chromosomes 1, 2, 3, 4, 5, 7, 26, 35, and 40, which were close to *EP300*,*ENSAPLG00020000057*, *GBE1*,*SPRY2*,*NDFIP2*,*BCKDHB*,*IL1RAP*,*ENSAPLG00020011494*,*Mdga2*,*RNLS*, *ENSAPLG00020005166*, *FOXP4*, *ENSAPLG00020010580*, and *ENSAPLG00020009757*, were found to be associated with weight at 42 days of age (Appendix A).

### 3.4. Functional Analysis

To better understand the function of the identified candidate genes related to the seven FE measures, KEGG and GO enrichment analyses were performed. For the weight at 21 days trait, the candidate genes were mostly enriched in the ribosome pathway, structural constituent of ribosome, large ribosomal subunit, ribosome biogenesis, translation, and nucleoplasm (Appendix A). Moreover, the terms phosphatidylinositol dephosphorylation, embryonic skeletal system morphogenesis, ATP binding, zinc ion binding, inositol phosphate metabolism pathways, phosphatidylinositol signaling system, and vascular smooth muscle contraction were enriched by candidate genes related to the BWG trait (Appendix A). Furthermore, the candidate genes related to the weight at 42 days trait were involved in the Transforming growth factor beta (TGF-β) signaling pathway, Notch signaling pathway, Wnt signaling pathway, FoxO signaling pathway, negative regulation of gluconeogenesis terms, fat cell differentiation terms, hydrolase activity, and on ester bond epinephrine binding terms (Appendix A). Pathways of starch and sucrose metabolism, galactose metabolism, metabolism, carbohydrate metabolic process, integral component of membrane, and carbohydrate-binding terms were enriched by the FI-related candidate genes (Figure 4a). The pathways of drug metabolism (such as cytochrome P450), metabolism of xenobiotics by cytochrome P450, glutathione metabolism, drug metabolism of other enzymes, glycosaminoglycan biosynthesis—keratan sulfate, glycosphingolipid biosynthesislacto and neolacto series, TGF-β signaling pathway, and the terms of GTPase activator activity, positive regulation of GTPase activity, ventricular cardiac muscle cell development, growth cone membrane, positive regulation of myoblast differentiation, and pituitary gland development were involved in RFI-associated genes (Figure 4b). In addition, the FCR-associated genes were involved in metabolic pathways, oxidative phosphorylation, cardiac muscle contraction, the Fanconi anemia pathway, and nucleotide excision repair pathways (Figure 5).

## 4. Discussion

FE is an important economic trait for farm animals; thus, the identification of major genes related to FE measures is of great interest. Some variants, genes, and pathways are considered to be related to FE traits in poultry and livestock. However, the FE trait is a complex phenotype that can be difficult to assess as compared with other traits [33]. Some signaling pathways have been related to FE, but very few candidate genes have been identified [34,35,36]. One possible explanation for this lack of knowledge is that FE is a complicated economic trait that may be controlled by many different genes. However, each gene may not have a significant effect on the functioning of the body. Previous studies of feed traits and growth in Pekin duck found SNP between the fourth and fifth exon of the *IL1RAPL1* gene could explain 2.5% of FCR phenotypic variation [37]. Some studies about FE evidence show that some cytokines related to immune response have been found to locate within the FE QTL regions. In chicken, some interleukins (IL10, IL7R) are associated with growth and gut length [38]. Cobb chickens and Beijing-You chickens examined here showed some unique characteristics, and only 127 genes associated with RFI were identified in both breeds [39]. In addition, 14 differently expressed lincRNAs were found in the high- and low-FE pigs [40,41,42]. In ducks, previous studies of feed traits and growth in Pekin duck found SNP between the fourth and fifth exon of the *IL1RAPL1* gene could explain 2.5% of FCR phenotypic variation [35]. Moreover, few studies have focused on exploring the underlying mechanisms of duck FE, and those that have often comprise smaller sample sizes due the high costs and labor required associated with it.

GWAS and quantitative trait locus mapping have been performed in an increasing number of studies to identify candidate variants of FE traits. However, the main genes related to FCR remain unclear and only a few candidate variants of duck FE have been identified to date [7]. In the present study, descriptive statistics were used to explore the correlation between seven FE traits. Overall, all traits showed significant positive correlations of varying degrees. Furthermore, detailed genetic sequencing revealed six candidate SNPs that were associated with BWG. Functional analysis also showed that the BWG-related genes were mainly involved in phosphatidylinositol dephosphorylation, embryonic skeletal system morphogenesis, ATP binding, zinc ion binding, pathways of inositol phosphate metabolism, phosphatidylinositol signaling system, and vascular smooth muscle contraction. Concerning FCR, the identified related candidate genes were involved in the metabolic pathway, whereas those related to the weight at 42 days trait were involved in the TGF-β signaling pathway, Notch signaling pathway, Wnt signaling pathway, and FoxO signaling pathway. Noteworthy, after 42 days, ducks are in the fast growth stage. In addition, one of the most potent inhibitors of muscle growth, *MSTN*, is activated via the phosphorylation of Smad2/3 [7]. Lastly, genes related to FI were found to be more involved in the starch and sucrose metabolism, galactose metabolism, carbohydrate metabolism, and carbohydrate-binding pathways. The identification of associated SNPs represented a key pace forward in dissecting the genetic basis of FE-related traits in ducks, which were also helpful for further demonstrating molecular mechanisms of related traits and designing better selection methods for these traits.

## 5. Conclusions

Taken together, this study has shown that 4 (FCR), 3 (FI), 36 (RFI), 6 (BWG), 8 (BW21), and 10 (BW42) candidate SNPs are associated with the BWG, FI, RFI, FCR, and weight at 21- and 42-days traits, respectively. These findings improve our understanding of poultry genetics and provide a basis for breeding programs aimed at maximizing the economic potential of duck breeding and farming.

## Figures and Tables

**Figure 1 animals-12-01532-f001:**
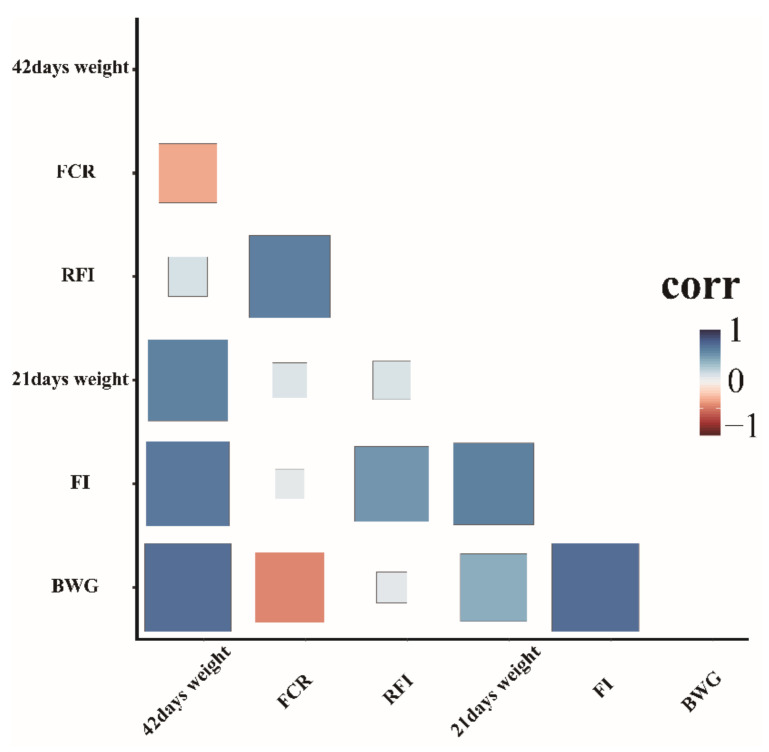
Pairwise Pearson correlation coefficients for the different FE traits analyzed.

**Figure 2 animals-12-01532-f002:**
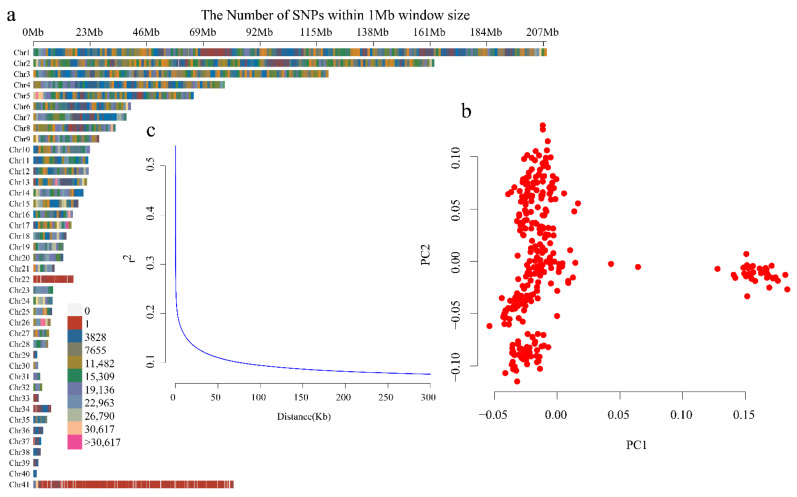
Single nucleotide polymorphism (SNP) distribution, study population structure, and linkage distribution (LD) decay. (**a**) SNP distribution in the chromosomes. (**b**) Principal component analysis (PCA) of all samples (*n* = 308). (**c**) LD decay analysis.

**Figure 3 animals-12-01532-f003:**
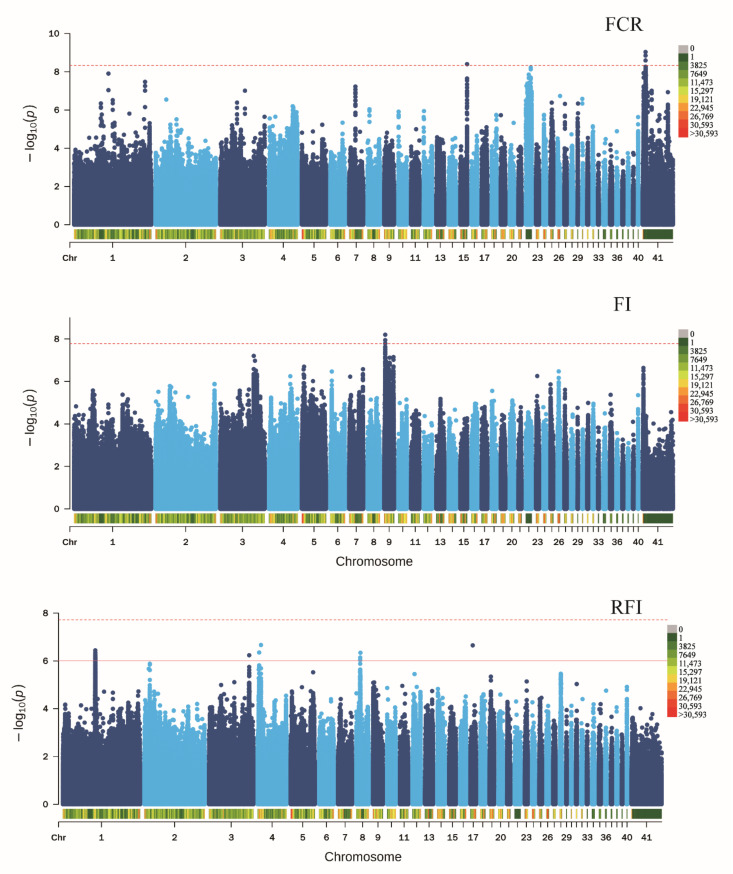
Genome-wide association study of the three FE traits in ducks. Manhattan plots in which the genomic coordinates of the SNPs are displayed along the horizontal axis, the negative logarithm of the association *p*-value for each SNP is displayed on the vertical axis. The red line indicates the significance threshold level after Bonferroni correction.

**Figure 4 animals-12-01532-f004:**
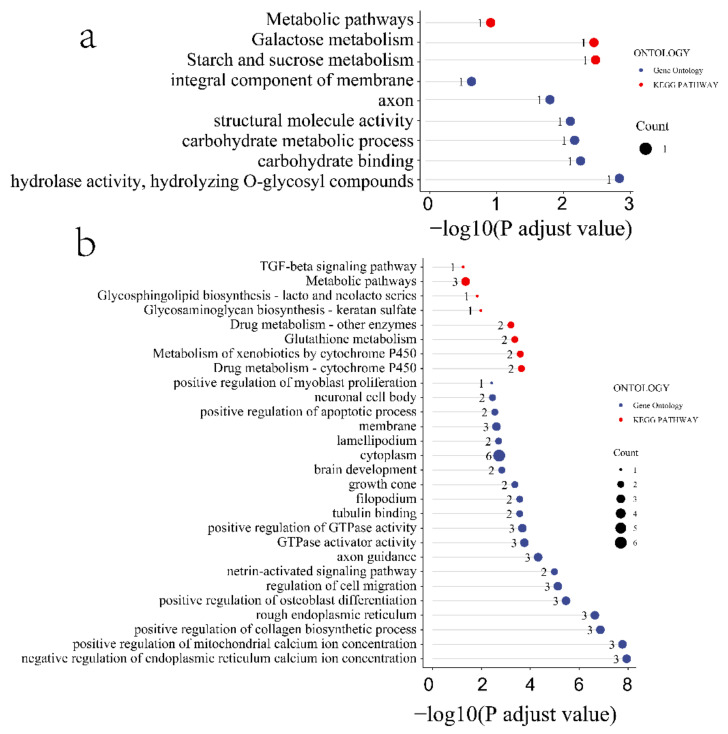
Functional enrichment analysis of the FE trait-related candidate genes. (**a**) Feed intake, (**b**) residual feed intake. Red and blue colored ribbons represent GO terms and KEGG pathways, respectively.

**Figure 5 animals-12-01532-f005:**
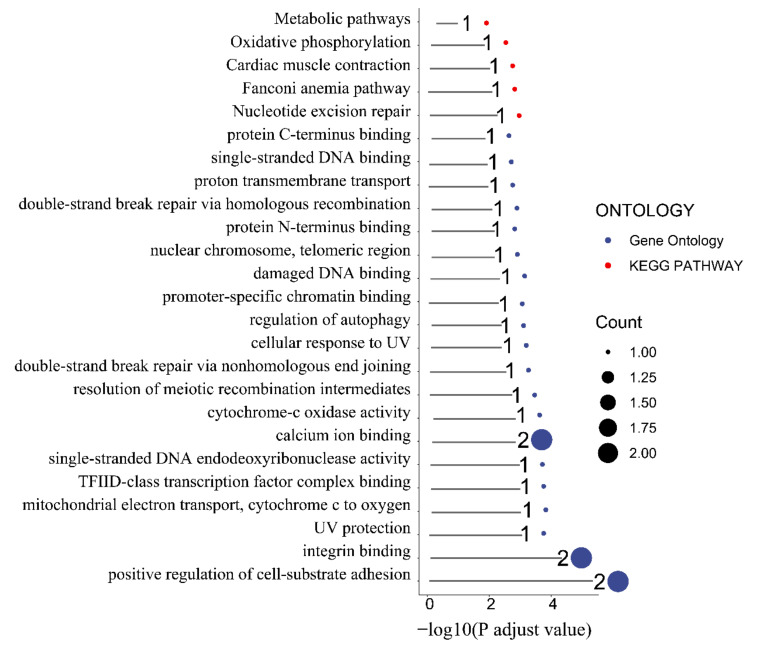
Functional enrichment analysis of the feed conversion ratio. Red and blue colored ribbons represent GO terms and KEGG pathways, respectively.

**Table 1 animals-12-01532-t001:** Descriptive statistics of FE and related traits.

Trait ^a^	Mean (g)	SD (g)	CV ^b^	Min (g)	Max (g)
42 days BW (g)	2351.83	314.59	0.13	1581	3074
FCR (g:g)	2.77	0.34	0.12	1.77	3.83
RFI (g/d)	24.15	310.86	−12.87	−924.3	941.78
21 days BW (g)	989.61	145.41	0.15	314	1345
FI (g/d)	3735.72	519.15	0.14	2444	5200
BWG (g)	1362.21	226.32	0.17	722	1932

^a^ FI, RFI, FCR, BWG, and BW represent daily feed intake, residual feed intake, feed conversion ratio, body weight gain, and body weight, respectively. ^b^ CV represents the coefficient of variation.

## Data Availability

The genome assembly and all of the re-sequencing data used in this research are deposited in the Genome Sequence Archive (GSA) at National Genomics Data Center (http://bigd.big.ac.cn/) Beijing Institute of Genomics, Chinese Academy of Sciences (GSA: CRA005019).

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
