# Peer review of "Genome-Wide Association Study of Feed Efficiency Related Traits in Ducks"

_animals, 2022, doi:10.3390/ani12121532_

Round 1
Reviewer 1 Report
Dear Authors,
thank you for addressing my suggestion in the revised version of your manuscript.
Kind regards
Author Response
Thank you for your comments on our manuscript
Reviewer 2 Report
Authors have incorporated the suggestions according to my previous comments improving the quality of the text.
Author Response
Thank you for your comments on our manuscript.
Reviewer 3 Report
Guo et al. performed the GWAS for feed efficiency traits which are key phenotypes for the livestock industry. The authors used the whole genome sequence data which is a very promising approach, although the sample size is relatively small.
My first concern about the manuscript is the unclear approach for modeling RFI, it is not clear how the authors measure Daily feed intake. Moreover, the middle metabolic weight should not be used as in equation 1 from the manuscript. I also did not understand why the author used Bodyweight gain instead of Average daily gain.
Secondly, the author used FarmCPU, it does not really benefit compared to other traditional approaches for GWAS in livestock.
Thirdly, the quality of the figure is too poor, I could not see or read anything in the current version.
Expected feed intake is not a real trait, the authors should not use it.
Line 14-16: It is not clear what is the difference between the body weight gain and body weight at 21 and 42 days. How many times did the authors measure the daily feed intake and body weight?
Line 16: “Overall, 6, 6, 4, 36, 36, 8, and 10 candidates single nucleotide” : the authors should mention these numbers belonging to which traits as lines 29-31, otherwise remove them or summarize them as total numbers of significant SNPs.
Line 18-19: For most of the pathways related to metabolism, the authors should specify which types of metabolism.
Line 30: Why did the authors map for expected daily feed intake, it does not make sense.
Line 30: body weight gain, did the authors mean Average daily gain. Specify the time frame of calculate the gain.
Line 32: List the key pathways
Line 53-55: it is not clear; the authors should mention the favorable genetic correlation between RFI and DFI.
Line 56: add more references for different livestock to support the statement.
Line 65-68: Please summarise the GWAS have been done for RFI and FCR to extend the introduction.
Line 76: Add the ethnic approval for the study.
How did the authors measure the feed intake, which is unclear from the manuscript? Line 94: what did the authors do for the calculation of FCR?
Where did the authors calculate expected feed intake?
Please check equation 1, how to write the mean and the fixed effects in the models.
The methods for calculation of RFI are not common, in the models if the first one should be average daily gain and the second should be the middle metabolic weight if the authors do not have any other phenotypes such as backfat or lean meat percentage. Except for the previous work from the authors, could the authors please provide more supporting references for modeling RFI?
Line 142: For the GWAS methods: I did not recommend FarmCPU over other more popular approaches such as EMAX or GENABEL, but it is fine in the statistical approach.
The quality of figures is too poor, the authors could move some figures of qqplot to supplementary files.
Table 1 and whole manuscript: The authors should remove EFI, it does not have any meaning for the FE traits.
Line 348: Remove and et.al... and change to .etc
Line 346-354: I did not know why the authors needed to add some information for miRNAs,
The discussion should focus on the potential roles of candidate genes as well as the mechanism influencing the feed inefficiency traits, and what could make the genetic differences between RFI and FCR.
Line 361-370: Why did the authors focus on BWG, are similar genetic markers found for Average daily gains in chicken or other species?
Author Response
Guo et al. performed the GWAS for feed efficiency traits which are key phenotypes for the livestock industry. The authors used the whole genome sequence data which is a very promising approach, although the sample size is relatively small.
Response: Thank you for reviewing our manuscript and providing suggestions to improve it. We have revised the manuscript accordingly and have responded to your comments point by point.
My first concern about the manuscript is the unclear approach for modeling RFI, it is not clear how the authors measure Daily feed intake. Moreover, the middle metabolic weight should not be used as in equation 1 from the manuscript. I also did not understand why the author used Bodyweight gain instead of Average daily gain.
Response: We would like to thank you for your careful reading, helpful comments, and constructive suggestions, which has significantly improved the presentation of our manuscript. We thank the reviewer for pointing out this issue. For daily feed intake measurements, we weighed each duck's initial feed addition and final remaining feed because we housed each duck in a single cage. Finally, the average daily feed intake was obtained by dividing the experimental days by the total feed intake. For your comment about the relevant parameters in Equation 1, such as intermediate metabolic weight and weight gain. Regarding the formula of RFI, we mainly refer to the relevant calculations of the following two papers[1, 2]. The deduction process of the RFI formula is described in detail in these two reports.
- Zhang, Y.-S., et al., Relationship between Residual Feed Intake and Production Traits in a Population of F(2) Ducks. The journal of poultry science, 2019. 56(1): p. 27-31.
- Zhang, Y., et al., Genetic parameters for residual feed intake in a random population of Pekin duck. Asian-Australas J Anim Sci, 2017. 30(2): p. 167-170.
Secondly, the author used FarmCPU, it does not really benefit compared to other traditional approaches for GWAS in livestock.
Response: Thank you for your comments. In fact, we also tried other methods to analyze our data. According to the results, we chose FarmCPU to analyze our data. At the same time, we found that there are many literatures that also analyze livestock and poultry data through FarmCPU. Below are some of those references[3-6].
- Abdalla, I.M., et al., Genome-Wide Association Study Identifies Candidate Genes Associated with Feet and Leg Conformation Traits in Chinese Holstein Cattle. Animals (Basel), 2021. 11(8).
- Gao, G., et al., Genome-Wide Association Study of Meat Quality Traits in a Three-Way Crossbred Commercial Pig Population. Front Genet, 2021. 12: p. 614087.
- Zhou, J., et al., Genome-wide association study of milk and reproductive traits in dual-purpose Xinjiang Brown cattle. BMC Genomics, 2019. 20(1): p. 827.
- Liu, L., et al., GWAS-Based Identification of New Loci for Milk Yield, Fat, and Protein in Holstein Cattle. Animals (Basel), 2020. 10(11).
Thirdly, the quality of the figure is too poor, I could not see or read anything in the current version.
Response: Thank you for the above suggestions. We have improved the quality of the all figure.
Expected feed intake is not a real trait, the authors should not use it.
Response: Thank you for your rigorous comment. We have removed the relevant content about expected feed intake trait in whole manuscript.
Line 14-16: It is not clear what is the difference between the body weight gain and body weight at 21 and 42 days. How many times did the authors measure the daily feed intake and body weight?
Response: Thank you for your rigorous comment. 21-day body weight refers to the duck's weight at 21 days of age, 42-day body weight refers to the duck's weight at 42 days of age, and body weight gain refers to the 42-day weight minus the 21-day weight. In addition, in order to reduce the experimental error caused by factors such as weighing, this study only carried out body weighing twice during the experimental period, 21 days and 42 days respectively. For the feed intake, 2 kg of feed was added every two days during the experiment, and the feed was ad libitum until the end of the experiment, and the remaining feed was weighed to obtain the average daily feed intake, which was the daily feed intake.
Line 16: “Overall, 6, 6, 4, 36, 36, 8, and 10 candidates single nucleotide”: the authors should mention these numbers belonging to which traits as lines 29-31, otherwise remove them or summarize them as total numbers of significant SNPs.
Response: We are grateful for the suggestion. According to your suggestion, we have mention the significant SNP number to belonging to which traits.
Line 18-19: For most of the pathways related to metabolism, the authors should specify which types of metabolism.
Response: Thanks for your comment. We have added specific metabolic pathways where appropriate.
Line 30: Why did the authors map for expected daily feed intake, it does not make sense.\
Response: Thank you for your rigorous comment. We have removed the relevant content about expected feed intake trait in whole manuscript.
Line 30: body weight gain, did the authors mean Average daily gain. Specify the time frame of calculate the gain.
Response: Thanks for your comment, and the Body weight gain refers to the amount of body weight gain during the whole experimental period. And we have added the information which including the specify the time frame of calculate the gain.
Line 32: List the key pathways
Response: Thanks for your comment. We have added specific metabolic pathways where appropriate.
Line 53-55: it is not clear; the authors should mention the favorable genetic correlation between RFI and DFI.
Response: Thanks for your comment. As your comment, we thought you might be wondering about genetic correlations between RFI and daily feed intake. We have cited some reference reported the genetic correlation between RFI and DFI of sheep and turkey.
Line 56: add more references for different livestock to support the statement.
Response: Thanks for your comment. We have added more references about RFI in different livestock.
Line 65-68: Please summarise the GWAS have been done for RFI and FCR to extend the introduction.
Response: Thanks for your comment. We have added some summaries on the application of GWAS in identifying candidate mutations in RFI and FCR.
Line 76: Add the ethnic approval for the study.
Response: We are grateful for the suggestion, and we have added the ethnic approval for the study.
How did the authors measure the feed intake, which is unclear from the manuscript? Line 94: what did the authors do for the calculation of FCR?
Response: Thanks for your rigorous comment. For measurement the feed intake, 2 kg of feed was added every two days during the experiment, and the feed was ad libitum until the end of the experiment, and the remaining feed was weighed to obtain the feed intake throughout the experimental period. Calculate FCR by the following formula.
Where did the authors calculate expected feed intake?
Response: Thanks for your rigorous comment. We have removed the relevant content about expected feed intake trait in whole manuscript.
Please check equation 1, how to write the mean and the fixed effects in the models.
Response: Thanks for your comment. We have re-check the all equation and change the mistake of fixed effects in the models.
The methods for calculation of RFI are not common, in the models if the first one should be average daily gain and the second should be the middle metabolic weight if the authors do not have any other phenotypes such as backfat or lean meat percentage. Except for the previous work from the authors, could the authors please provide more supporting references for modeling RFI?
Response: Regarding the formula of RFI, we mainly refer to the relevant calculations of the following two papers[1, 2]. The deduction process of the RFI formula is described in detail in these two reports.
- Zhang, Y.-S., et al., Relationship between Residual Feed Intake and Production Traits in a Population of F(2) Ducks. The journal of poultry science, 2019. 56(1): p. 27-31.
- Zhang, Y., et al., Genetic parameters for residual feed intake in a random population of Pekin duck. Asian-Australas J Anim Sci, 2017. 30(2): p. 167-170.
Line 142: For the GWAS methods: I did not recommend FarmCPU over other more popular approaches such as EMAX or GENABEL, but it is fine in the statistical approach.
Response: Thank you for your comments. In fact, we also tried other methods to analyze our data. According to the results, we chose FarmCPU to analyze our data. At the same time, we found that there are many literatures that also analyze livestock and poultry data through FarmCPU. Below are some of those references[3-6].
- Abdalla, I.M., et al., Genome-Wide Association Study Identifies Candidate Genes Associated with Feet and Leg Conformation Traits in Chinese Holstein Cattle. Animals (Basel), 2021. 11(8).
- Gao, G., et al., Genome-Wide Association Study of Meat Quality Traits in a Three-Way Crossbred Commercial Pig Population. Front Genet, 2021. 12: p. 614087.
- Zhou, J., et al., Genome-wide association study of milk and reproductive traits in dual-purpose Xinjiang Brown cattle. BMC Genomics, 2019. 20(1): p. 827.
- Liu, L., et al., GWAS-Based Identification of New Loci for Milk Yield, Fat, and Protein in Holstein Cattle. Animals (Basel), 2020. 10(11).
The quality of figures is too poor, the authors could move some figures of qqplot to supplementary files.
Response: Thank you for the above suggestions. We have improved the quality of the all figure.
Table 1 and whole manuscript: The authors should remove EFI, it does not have any meaning for the FE traits.
Response: Thanks for your rigorous comment. We have removed the relevant content about expected feed intake trait in whole manuscript.
Line 348: Remove and et.al.... and change to .etc
Response: Thank you for your comments. We have changed the et.al to .etc in manuscript.
Line 346-354: I did not know why the authors needed to add some information for miRNAs,
The discussion should focus on the potential roles of candidate genes as well as the mechanism influencing the feed inefficiency traits, and what could make the genetic differences between RFI and FCR.
Response: Thank you for your comments. We have remove the information for miRNAs, and re-write some paragraph in part of discussion.
Line 361-370: Why did the authors focus on BWG, are similar genetic markers found for Average daily gains in chicken or other species?
Response: Thank you for your comments. Cause of the equation 1 which the equation of RFI, we focus on the BWG.

Round 2
Reviewer 3 Report
The authors have addressed most of my comments. However, there are still points to clarify from the current version.
“candidate single nucleotide polymorphisms were found to be associated” change to single nucleotide polymorphisms were significantly associated with ….
Make the same change in line 30;
“intermediate between daily feed intake (DFI) and RFI was”: what did the authors mean intermediate, the genetic or phenotypic correlation.
Line 54 -57: Might better give the example with chicken, it is close species to the duck.
Line 66: Abbreviation for GWAS should be defined here.
Line 71-72: the authors still need a reference for it.
Line 72-74: Should specify for which species, there are many for pigs and cattle or even chicken.
Line 103-107: Still the authors did not provide the information about how the feed intake was recorded in the manuscript.
Line 110: b1 and b2: 1 and 2 should be used in under script form
bonferroni should change to Bonferroni
Line 187: Where did the authors get an additive genetic variance.
The quality of figure 1 does not improve, still could not see anything from the current version. The authors might. The authors might consider the distribution and the qq plots as supplementary files.
The quality of figures 2, 3, and 4 is poor as well. The authors might separate them into small figures. Or just keep the main figure for RFI, DFI, and FCR as they are key traits. Otherwise, the authors should keep at least one table. I recommend that the authors list the SNPs and genes for RFI, FCR, DFI, and BWG in table 2.
Line 312: it depends on traits, it is not true for all traits. Some health traits might be harder to measure than FE.
Line 312-314: What species did the authors mention. I believe at least for pigs and cattle many genes and SNPs have been identified for FE-related traits.
Line 318 and some other lines. Please use FE instead of feed efficiency.
Line 317-318, 325-326: The information is repeated.
Line 311-318: Are any genes listed here found in the current study.
Author Response
The authors have addressed most of my comments. However, there are still points to clarify from the current version.
Response: Thank you for reviewing our manuscript and providing suggestions to improve it. We have revised the manuscript accordingly and have responded to your comments point by point.
“candidate single nucleotide polymorphisms were found to be associated” change to single nucleotide polymorphisms were significantly associated with ….
Make the same change in line 30;
Response: Thanks for your comment. And we have change it in the manuscript.
“intermediate between daily feed intake (DFI) and RFI was”: what did the authors mean intermediate, the genetic or phenotypic correlation.
Response: Thanks for your comment. The sentence of “intermediate between daily feed intake (DFI) and RFI” means genetics correlation between daily feed intake (DFI) and RFI.
Line 54 -57: Might better give the example with chicken, it is close species to the duck.
Response: Thanks for your comment. We have cited the reference of chicken in manuscript.
Line 66: Abbreviation for GWAS should be defined here.
Response: Thanks for your comment. And we have change it in the manuscript.
Line 71-72: the authors still need a reference for it.
Response: Thanks for your comment. We have cited the reference of chicken in manuscript.
Line 72-74: Should specify for which species, there are many for pigs and cattle or even chicken.
Response: Thanks for your comment. We have cited the reference of pigs and cattle or even chicken in manuscript.
Line 103-107: Still the authors did not provide the information about how the feed intake was recorded in the manuscript.
Response: Thanks for your comment. We have provided the information about feed intake in manuscript.
Line 110: b1 and b2: 1 and 2 should be used in under script form
bonferroni should change to Bonferroni
Response: Thanks for your comment. And we have change it in the manuscript.
Line 187: Where did the authors get an additive genetic variance.
Response: Thanks for your comment. We have removed the additive genetic variance relate sentence.
The quality of figure 1 does not improve, still could not see anything from the current version. The authors might. The authors might consider the distribution and the qq plots as supplementary files.
The quality of figures 2, 3, and 4 is poor as well. The authors might separate them into small figures. Or just keep the main figure for RFI, DFI, and FCR as they are key traits. Otherwise, the authors should keep at least one table. I recommend that the authors list the SNPs and genes for RFI, FCR, DFI, and BWG in table 2.
Response: Thanks for your comment. We have improved the quality of the images in the article. At the same time according to your suggestion, we will add the relevant pictures of phenotypes other than RFI, FI and FCR to the supplementary files.
Line 312: it depends on traits, it is not true for all traits. Some health traits might be harder to measure than FE.
Response: Thanks for your comment. And we have change it in the manuscript.
Line 312-314: What species did the authors mention. I believe at least for pigs and cattle many genes and SNPs have been identified for FE-related traits.
Response: Thanks for your comment. We have cited the reference of pigs and cattle or chicken FE related traits in manuscript.
Line 318 and some other lines. Please use FE instead of feed efficiency.
Response: Thanks for your comment. And we have change it in the manuscript.
Line 317-318, 325-326: The information is repeated.
Response: Thanks for your comment. And we have removed it in the manuscript.
Line 311-318: Are any genes listed here found in the current study.
Response: Thanks for your comment. We didn’t find any recent studies that were similar to the genes in my study. We speculate that this may be due to differences in species.

Round 3
Reviewer 3 Report
The authors have addressed my comments. However, the quality of the figures is not improved.
I could not read the values in Figures 1 and 3 while the labels of terms in figure 4 are not clear.
Figure 1 does not have much information, The authors could just write it down. Or using only the number of color codes, but not both.
Why did not the authors report the CV for RFI in table 1
Author Response
The authors have addressed my comments. However, the quality of the figures is not improved.
Response: Thank you for reviewing our manuscript and providing suggestions to improve it. We have revised the manuscript accordingly and have responded to your comments point by point.
I could not read the values in Figures 1 and 3 while the labels of terms in figure 4 are not clear.
Response: Thank you for the above suggestions. We have improved the quality of the all figures.
Figure 1 does not have much information. The authors could just write it down. Or using only the number of color codes, but not both.
Response: Thank you for the above suggestions. We have change it.
Why did not the authors report the CV for RFI in table 1.
Response: Thank you for the above suggestions. We have added it.

This manuscript is a resubmission of an earlier submission. The following is a list of the peer review reports and author responses from that submission.
Round 1
Reviewer 1 Report
Dear Authors,
I reviewed your manuscript titled “Genome-wide association study of feed efficiency traits in ducks” and I just have some requests to improve it.
You use often acronyms, but you should explain it the first time you use. (i.e. line 254: KEGG and GO; line 260: TGF)
The AA should add the references for the software Plink and for the packages of R used (i.e. ggplt2, Johnson, etc)
Line 175: where is “dj” in the equation?
Table1:
- Number of decimals should be constant
- are the values of RFI correct?
- What is CV (coefficient of variation)? In the notes it should be an explanation of the acronym used in the table
- I suggest to delete line 200 “an=308”
Finally, all figures have a low quality resulting not readable, you should improve the quality of each picture.
Sincerely
Reviewer 2 Report
- This study entitled ‘Genome-wide association study of feed efficiency traits in ducks’ is interesting to the reader. However, I felt that this study needs to be clearer and change all of the figures. Because the quality of all the figures is too low to be review, I recommend to the authors resubmit it after editing the figure.
- Please check the manuscript as it contains some sentences similar to a specific thesis.
‘Genome-wide association study reveals putative role of gga-mi R-15a in controlling feed conversion ratio in layer chickens. 2017.’
- Describe the method and age for get all the phenotype data. (such as the content such as period of BWG, FCR, FI etc.)
- All factors expressed in the GWAS formula should be fully explained, and factors not provided in the formula or not used in the calculation should not be written (L175 dj, formular (4) effect, ??2)
- The first paragraph of the discussion is suitable for the introduction.
- It would be a better manuscript if more considerations on the results of this study were added to the discussion section. (ex. The explanartion and cause of the relationship between each functional analysis result and the phenotypes)
- In conclusion part, two sentences are almost same with ‘L29-31’, and ‘L19-21’, respectively.
Reviewer 3 Report
Guo et al in their manuscript try to shed light in a difficult trait, that of feed efficiency, in ducks using a GWA study. Authors offer valuable novel information especially in a poultry specie not so well explores like chickens. The work is well presented, however there are some points to be clarified or corrected appropriately as follows:
l.34 & 76 : 308 samples. Were the samples related or unrelated. Did they selected at random? Please mention.
Materials and methods section. a) Please specify the diet that animals were fed and how the quantity was re-adapted in during the whole experimental period? i.e. Was it stable during the experiment? What feed management was implemented (feed in lots or for each animal?) In addition, how expected feed intake was estimated? b) Could you please specify the sex of the animals that used in the genetic analyses and if both sexes were used did any differences between sexes were observed in the involved genes?
l.180-186. Please specify the thresholds that were used in the analysis.
l 112 Add reference for the used protocol
190-191 The values are for both time points (as a mean)? In the table there is no -924.352 value.
Figures 2-3-4 are not easily readable. Please improve quality.
Discussion. Authors may improve the discussion section trying to make a short connection of the identified genes with metabolism and to which direction (positive or negative) these genes influence the FE, borrowing some examples from other livestock species in case that experiments are not available for duck or other poultry species.